

# Effects of elevated pCO$_2$ and nutrient enrichment on the growth, photosynthesis, and biochemical compositions of the brown alga *Saccharina japonica* (Laminariaceae, Phaeophyta)

Yaoyao Chu[1], Yan Liu[1,2], Jingyu Li[1,2] and Qingli Gong[1,2]

[1] College of Fisheries, Ocean University of China, Qingdao, Shandong, China
[2] The Key Laboratory of Mariculture (Ocean University of China), Ministry of Education, Qingdao, Shandong, China

## ABSTRACT

Ocean acidification and eutrophication are two major environmental issues affecting kelp mariculture. In this study, the growth, photosynthesis, and biochemical compositions of adult sporophytes of *Saccharina japonica* were evaluated at different levels of pCO$_2$ (400 and 800 µatm) and nutrients (nutrient-enriched and non-enriched seawater). The relative growth rate (RGR), net photosynthetic rate, and all tested biochemical contents (including chlorophyll (Chl) *a*, Chl *c*, soluble carbohydrates, and soluble proteins) were significantly lower at 800 µatm than at 400 µatm pCO$_2$. The RGR and the contents of Chl *a* and soluble proteins were significantly higher under nutrient-enriched conditions than under non-enriched conditions. Moreover, the negative effects of the elevated pCO$_2$ level on the RGR, net photosynthetic rate, Chl *c* and the soluble carbohydrates and proteins contents were synergized by the elevated nutrient availability. These results implied that increased pCO$_2$ could suppress the growth and biochemical composition of adult sporophytes of *S. japonica*. The interactive effects of ocean acidification and eutrophication constitute a great threat to the cultivation of *S. japonica* due to growth inhibition and a reduction in quality.

# INTRODUCTION

Due to intensive anthropogenic activities in recent years, the level of CO$_2$ in the atmosphere has increased from 285 µatm in 1975 to >400 µatm at present (*IPCC, 2014*). This caused an increase in the concentration of dissolved CO$_2$ in the ocean, resulting in a decrease in seawater pH, which is called ocean acidification (OA). Because of OA, the seawater carbonate system has also changed, affecting the physiological performances of marine organisms, species interactions, and coastal ecosystems (*Connell & Russell, 2010*;

Corresponding authors
Yan Liu, qd_liuyan@ouc.edu.cn
Qingli Gong, qingli@vip.sina.com

Enochs et al., 2015; Myers et al., 2017; Ullah et al., 2018). On the other hand, due to a large amount of wastewater emissions derived from industrial and agricultural production, eutrophication has been another important environmental issue affecting coastal waters worldwide (Geertz-Hansen et al., 1993; Smith et al., 2003). Excessive nutrient inputs can lead to harmful algal blooms (Bricker et al., 2008; Cai et al., 2011; Smetacek & Zingone, 2013), which decrease oxygen concentrations in the water column and produce toxins that might be lethal to marine organisms, thereby resulting in shifts in species dominance and community structure (Glibert et al., 2005; Xiao et al., 2017).

Kelp are the dominant taxa in the sublittoral zones of temperate and polar coastal regions worldwide and provide habitat and nursery ground for a large number of marine organisms (Steneck et al., 2002; Graham, 2004). Due to their ecological importance, many studies have focused on how environmental changes, such as OA and eutrophication, affect the growth and physiology of kelp species (Gaitán-Espitia et al., 2014; Agatsuma et al., 2014; Oh, Yu & Choi, 2015; Gao et al., 2017c). Some results showed that OA negatively influenced the microscopic development of *Macrocystis pyrifera* and *Undaria pinnatifida* (Gaitán-Espitia et al., 2014; Xu et al., 2015; Gao et al., 2019a), which may be due to the inhibition of cell activities (Ragazzola et al., 2012). In contrast, kelp sporophytes exhibited positive physiological responses to OA (Schmid, Mills & Dring, 1996; Olischlaeger et al., 2012; Gao et al., 2019a; Gao et al., 2019b) due to stimulation of photosynthetic activities by increased inorganic carbon (Olischlaeger et al., 2012). Additionally, kelp species benefit from an increase in nutrient availability in seawater. For example, a nutrient supply increased the maturation of gametophytes and the growth and production of sporophytes (Mizuta, Narumi & Yamamoto, 2001; Agatsuma et al., 2014; Endo et al., 2017). Despite the important effects of OA and eutrophication on kelp species, very few studies have considered the combined effects of these factors on the physiological characteristics of kelp species.

The canopy-forming kelp *Saccharina japonica* Areschoug inhabits subtidal zones of northwestern Pacific countries, including Japan, Korea and China (Selivanova, Zhigadlova & Hansen, 2007; Liu et al., 2009). This kelp has also been commercially cultivated in these countries because adult sporophytes are used as food for humans and as raw industrial materials (Liu et al., 2012; Liu et al., 2014; Hwang et al., 2018). Previous studies on the physiological responses of *S. japonica* to OA focused mainly on the microscopic stages (Xu et al., 2015; Gao et al., 2019a) and juvenile sporophyte (Gao et al., 2019a). However, the responses of adult sporophytes to OA should also be investigated because many seaweeds showed different responses to OA during different developmental stages (Olischlaeger et al., 2012; Gao et al., 2018a; Gao et al., 2018b). OA could also significantly affect the biochemical composition of seaweeds, including protein and carbohydrates (Andria, Vergara & Perez-Llorens, 1999; Xu et al., 2017; Gao et al., 2018a), which is considered a criterion to measure the quality of cultivated algae. On the other hand, an increase in the nutrient concentration significantly enhanced the growth, photosynthetic activities, and nutrient uptake of *S. japonica* (Mizuta & Maita, 1991; Gao et al., 2017c). Nevertheless, very few studies have been conducted to evaluate the potential interactive effect of OA and eutrophication on the growth and quality of this kelp species.

Therefore, in the present study, we investigated the combined effects of OA and eutrophication on the growth, photosynthesis, and biochemical composition of adult sporophytes of *S. japonica*. According to previous studies, we hypothesized that OA would inhibit the growth and quality of this kelp and that excessive nutrients would affect its physiological responses to OA. The results of this study are expected to provide valuable information for improving the cultivation production of *S. japonica* in China.

## MATERIALS & METHODS

### Algal collection and maintenance

Adult sporophytes of *S. japonica* were collected from cultivated populations in Rongcheng, Shandong, China (36°07′N, 120°19′E), in May 2018. The samples were transported quickly to the laboratory using a cold plastic box filled with seawater within 5 h. Healthy sporophytes were selected and rinsed several times with sterilized seawater to remove epiphytic organisms and detritus. More than 60 discs (1.4 cm in diameter) were punched from the meristem with a cork borer for the subsequent experiments. The discs were stock-cultured in a plastic tank containing 6 L filtered seawater, which was obtained from the coast of Taipingjiao, Qingdao with a salinity of approximately 30 psu. These discs were kept at an irradiance of 180 $\mu$mol photons m$^{-2}$ s$^{-1}$, a 12:12 h light/dark cycle, and 10 °C, which was the seawater temperature of the collection area, for 3 days to reduce the negative effects of excision.

### Culture experiment and growth

A culture experiment was conducted over a period of 6 days under combinations of two pCO$_2$ levels (400 and 800 $\mu$atm) and two nutrient levels (non-enriched natural seawater and nutrient-enriched seawater). There was a total of 4 experimental treatments and each treatment had three replicates. During the experiment, a light/dark cycle of 12:12 h and an irradiance of 180 $\mu$mol photon m$^{-2}$ s$^{-1}$ were held constant. This experiment used 12 side-arm flasks, with each flask containing 500 mL of natural seawater or 50% PESI-enriched seawater (*Tatewaki, 1966*). In the natural seawater treatment, the culture medium contained 27 $\mu$M NO$_3^-$ and 2 $\mu$M H$_2$PO$_4^-$, and the cultures with 50% PESI-enriched seawater contained 437 $\mu$M NO$_3^-$ and 28 $\mu$M H$_2$PO$_4^-$. The elevated nutrient level was based on studies referring to eutrophication (*Ménesguen et al., 2018*; *Gao et al., 2019c*), and nutrient limiting did not occur at the applied nutrient level during the experiment based on a preliminary experiment. Four discs were put into each flask, which was then gently aerated. The culture medium was renewed every 3 days.

For the experimental treatment, two pCO$_2$ levels were maintained in two CO$_2$ incubators: 400 $\mu$atm (ambient air) and 800 $\mu$atm (elevated pCO$_2$). The CO$_2$ levels were automatically regulated in two incubators (GXZ-380C-C02, Jiangnan Instruments Factory, Ningbo, China) by controlling the flow of ambient air and pure CO$_2$ gas. Autoclaved natural seawater with the present pCO$_2$ level (approximately 400 $\mu$atm) was used as a control. A pH meter (Orion STAR A211; Thermo Scientific) was used to measure the pH value of the medium in each flask. Total alkalinity (TA) was measured using an automatic alkalinity titrator by Gran acidimetric titration (848MPT, Titrino). The samples used to measure the

TA were poisoned with a saturated solution of mercuric chloride after filtering through cellulose acetate membranes (0.22 $\mu$m). The seawater carbonate chemistry parameters were calculated based on the values of the pH, TA, salinity, nutrients, the equilibrium constants K1 and K2 for carbonic acid dissociation (*Roy et al., 1993*), and KB for boric acid (*Dickson, 1990*), using CO2SYS software (*Lewis & Wallace, 1998*).

At the end of the experiment, the fresh weights of the discs were measured after being blotted with tissue paper. The relative growth rate (RGR) of each replicate was calculated using the following formula:

$$\text{RGR}(\%\ \text{day}^{-1}) = 100 \ln (W_t/W_0)/t$$

where $W_0$ is the initial fresh weight, $W_t$ is the final fresh weight, and $t$ is the number of days.

## Photosynthesis measurements

After the culture experiment, the net photosynthetic rate ($P_n$) of the discs was obtained using a manual FireSting $O_2$ II oxygen meter (Firesting $O_2$, Pyro Science). After measuring the fresh weight, four discs were transferred to the oxygen electrode cuvette with 330 mL of medium from the culture flask. Then the medium was magnetically stirred during the measurement to ensure even diffusion of oxygen. The temperature and light conditions were the same as for the abovementioned culture experiment. Prior to the measurements, the samples were allowed to acclimate to the conditions in the cuvette for 5 min. The oxygen concentration in the medium was recorded every 1 min for 10 min. The $P_n$ was normalized to $\mu$mol $O_2$ g$^{-1}$ FW h$^{-1}$.

## Chl *a* and *c* measurements

Approximately 0.2 g (fresh weight) of the discs was used for the extraction of chlorophyll (Chl) *a* and *c*. The discs were placed in 2 mL dimethyl sulfoxide for 5 min, and the absorption of the supernatant was measured at 665, 631, 582 and 480 nm using an ultraviolet absorption spectrophotometer (U-2900, HITACHI, Tokyo, Japan). Next, the same discs were placed in 3 mL acetone for 2 h. Then, the supernatant was transferred into a 10 mL tube, and 1 mL methanol and 1 mL distilled water were added. The absorbance of the supernatant was measured at 664, 631, 581 and 470 nm using an ultraviolet absorption spectrophotometer. The contents of Chl *a* and *c* were calculated according to *Seely, Duncan & Vidaver (1972)*.

## Soluble carbohydrates and proteins

Fresh mass samples (0.1 g) were ground in 2 mL of distilled water and diluted to 10 mL after the addition of 2 mL $MgCO_3$ suspension liquid. The crude mixture was centrifuged with a table centrifuge at 4,000 rpm for 5 min at 4 °C. Then, 1 mL supernatant was transferred into a glass tube and diluted to 2 mL with distilled water, and 8 mL anthrone reagent was added. The mixture was bathed in boiled water for 10 min. After the reaction medium had cooled down, the absorption at 620 nm was recorded, and the mixture was standardized with distilled water. The content of soluble carbohydrates was determined by the anthrone reagent method with glucose standards according to *Tang, Gong & Chen (1985)*.
**Table 1** Parameters of the seawater carbonate system in different treatments. -familz.

| Nutrient | $pCO_2$ ($\mu$atm) | pH | DIC ($\mu$mol kg$^{-1}$) | TA ($\mu$mol kg$^{-1}$) | $CO_3^{2-}$ ($\mu$mol kg$^{-1}$) | $HCO_{3-}$ ($\mu$mol kg$^{-1}$) | $CO_2$ ($\mu$mol kg$^{-1}$) |
|---|---|---|---|---|---|---|---|
| Non-enriched | 400 | $8.09 \pm 0.02^a$ | $1783.21 \pm 17.17^a$ | $1927 \pm 4.16^a$ | $102.16 \pm 9.79^a$ | $1665.91 \pm 24.95^a$ | $15.14 \pm 2.01^a$ |
| | 800 | $7.78 \pm 0.01^b$ | $1886.17 \pm 9.29^b$ | $1936 \pm 6.03^a$ | $53.57 \pm 5.47^b$ | $1799.04 \pm 13.80^b$ | $33.55 \pm 0.90^b$ |
| Enriched | 400 | $8.07 \pm 0.07^a$ | $1776.66 \pm 16.03^a$ | $1930 \pm 5.51^a$ | $107.71 \pm 9.39^a$ | $1654.85 \pm 23.48^a$ | $14.10 \pm 1.92^a$ |
| | 800 | $7.79 \pm 0.03^b$ | $1882.36 \pm 11.88^b$ | $1941 \pm 4.58^a$ | $57.60 \pm 6.93^b$ | $1793.82 \pm 17.64^b$ | $30.93 \pm 1.17^b$ |

**Notes.**

DIC, dissolved inorganic carbon; TA, total alkalinity.

Data present means $\pm$ SD. Different letters indicate statistical differences ($p < 0.05$) among different experimental treatments. The unites for TA and carbonate chemistry parameters are $\mu$mol kg$^{-1}$. Different letters indicate statistical differences ($p < 0.05$) among different experimental treatments.

Fresh samples (0.1 g) were homogenized with a mortar and pestle and 5 mL of liquid nitrogen. The extract was centrifuged at 2,500 rpm for 10 min and then used to determine the content of soluble protein. The absorbance of the supernatant at 595 nm was recorded using an ultraviolet spectrophotometer. The soluble proteins of the alga was estimated using a combination of Coomassie Brilliant Blue G-250 dye and bovine albumin, as described by *Kochert (1978)*.

### Statistical analysis

All data in the present study are reported as the mean $\pm$ SD ($n = 3$). Prior to the analysis, tests for the normal distribution (Shapiro–Wilk test, $p > 0.05$) and homogeneity (Levene's test, $p > 0.05$) of variance were conducted. A two-way analysis of variance (ANOVA) was used to test the effects of the nutrient and $pCO_2$ levels on the RGR, net photosynthetic rate, and the contents of Chl $a$ and $c$ and soluble carbohydrates and protein. A Tukey HSD test was conducted to determine the significance levels of the factors ($p < 0.05$). All statistical analyses were performed using SPSS 22.0 software.

## RESULTS

### Seawater carbonate chemistry

The effects of $pCO_2$ and the nutrient level on the seawater carbonate parameters were detected (Table 1). The two-way ANOVA ($p = 0.05$) showed that $pCO_2$ had a significant effect on all parameters except for TA, whereas the nutrient level did not have a significant influence on any parameter. The Tukey HSD comparison ($p = 0.05$) showed that elevated $pCO_2$ decreased the pH value by 0.2 in both the enriched and non-enriched nutrient treatments and $CO_3^{2-}$ by 48% (non-enriched) and 47% (enriched), but it increased the dissolved inorganic carbon (DIC) by 6% (non-enriched) and 6% (enriched), $HCO_3^-$ by 8% (non-enriched) and 8% (enriched), and $CO_2$ by 122% (non-enriched) and 119% (enriched).

### Growth

The RGR values were significantly affected by $pCO_2$ and nutrients, both individually and interactively (Fig. 1; Table 2). At each $pCO_2$ level, the RGR values were significantly higher under the nutrient-enriched condition than under the non-enriched condition. Similarly, for both nutrient levels, the RGR was significantly lower at the higher $pCO_2$ level than

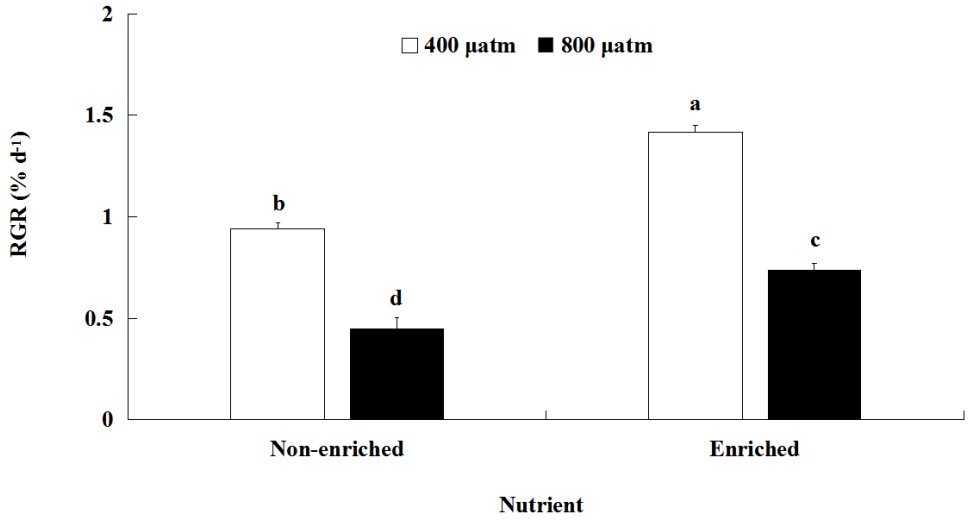

**Figure 1 Relative growth rate (RGR) of *S. japonica* cultured for 6 days under two pCO$_2$ and two nutrient levels.** Data represents mean $\pm$ SD ($n = 3$ replicates). Different letters indicate statistical differences ($p < 0.05$) among different experimental treatments.

**Table 2 Analysis of of two-way ANOVA showing the effects of pCO$_2$ and nutrient level and their interactions on the growth (RGR), photosynthesis, chlorophyll *a* and *c* (Chl *a* and Chl *c*), soluble carbohydrates and soluble proteins.**

| Parameter | Source of variation | df | Ms | F | p |
|---|---|---|---|---|---|
| Relative growth rate | pCO$_2$ | 1 | 1.025 | 687.175 | <0.001 |
| | Nutrient | 1 | 0.436 | 292.409 | <0.001 |
| | pCO$_2$ × Nutrient | 1 | 0.027 | 18.071 | 0.003 |
| Net photosynthetic rate | pCO$_2$ | 1 | 0.018 | 8.074 | 0.022 |
| | Nutrient | 1 | 0.003 | 1.527 | 0.252 |
| | pCO$_2$ × Nutrient | 1 | 0.006 | 2.881 | 0.128 |
| Chlorophyll *a* | pCO$_2$ | 1 | 0.028 | 55.091 | <0.001 |
| | Nutrient | 1 | 0.005 | 9.81 | 0.014 |
| | pCO$_2$ × Nutrient | 1 | 0.009 | 18.293 | 0.003 |
| Chlorophyll *c* | pCO$_2$ | 1 | 0.001 | 14.502 | 0.005 |
| | Nutrient | 1 | <0.001 | 0.036 | 0.854 |
| | pCO$_2$ × Nutrient | 1 | <0.001 | 4.823 | 0.059 |
| Soluble carbohydrates | pCO$_2$ | 1 | 0.878 | 8.192 | 0.021 |
| | Nutrient | 1 | 0.519 | 4.844 | 0.059 |
| | pCO$_2$ × Nutrient | 1 | 0.549 | 5.119 | 0.054 |
| Soluble proteins | pCO$_2$ | 1 | 4.971 | 20.186 | 0.002 |
| | Nutrient | 1 | 9.781 | 39.717 | <0.001 |
| | pCO$_2$ × Nutrient | 1 | 0.151 | 0.615 | 0.456 |

at the lower level. Furthermore, the difference in RGR between the two pCO$_2$ levels was further increased under nutrient-enriched conditions. The RGR showed a maximum of 1.417% day$^{-1}$ at the lower pCO$_2$ level and nutrient-enriched conditions.

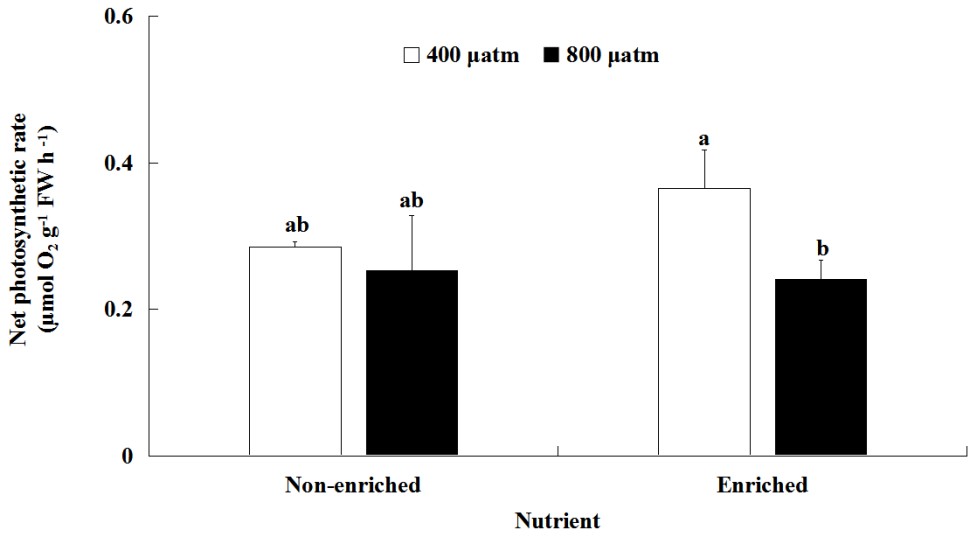

**Figure 2** **Net photosynthetic rate ($P_n$) of _S. japonica_ cultured for 6 days under two $pCO_2$ and two nutrient levels.** Data represents mean $\pm$ SD ($n = 3$ replicates). Different letters indicate statistical differences ($p < 0.05$) among different experimental treatments.

## Photosynthesis

The $P_n$ values were significantly different between the two $pCO_2$ levels (Fig. 2; Table 2). However, there was no significant effect of nutrients on $P_n$ values. A significant interaction between $pCO_2$ levels and nutrients was also not detected. Under nutrient-enriched conditions, the $P_n$ values were significantly higher at lower $pCO_2$ than at higher $pCO_2$. However, at both $pCO_2$ levels, the $P_n$ values did not significantly differ between the two nutrient levels. The $P_n$ values showed a maximum of 0.365 $\mu$mol $O_2$ $g^{-1}$ FW $h^{-1}$ under the lower $pCO_2$ level and nutrient-enriched conditions.

## Chl _a_ and _c_ contents

The Chl _a_ content was significantly affected by $pCO_2$ and nutrients, both individually and interactively (Fig. 3A; Table 2). Under non-enriched conditions, the Chl _a_ contents was significantly higher at lower $pCO_2$ than at higher $pCO_2$. However, under nutrient-enriched conditions, the Chl _a_ contents showed no significant difference between the two $pCO_2$ levels. Additionally, at higher $pCO_2$, the Chl _a_ contents was significantly higher under nutrient-enriched conditions than under non-enriched conditions. However, at lower $pCO_2$, the Chl _a_ content did not show significant differences between the two nutrient levels. The Chl _a_ content showed a minimum of 0.308 mg $g^{-1}$ at higher $pCO_2$ and non-enriched conditions.

The Chl _c_ content was significantly affected by $pCO_2$ levels (Fig. 3B; Table 2). There was no significant effect of nutrients on the Chl _c_ contents. Significant interaction between $pCO_2$ and nutrients was not detected. Under nutrient-enriched conditions, the Chl _c_ content was significantly higher at lower $pCO_2$ than at higher $pCO_2$. However, under non-enriched conditions, the Chl _c_ content showed no significant differences between the

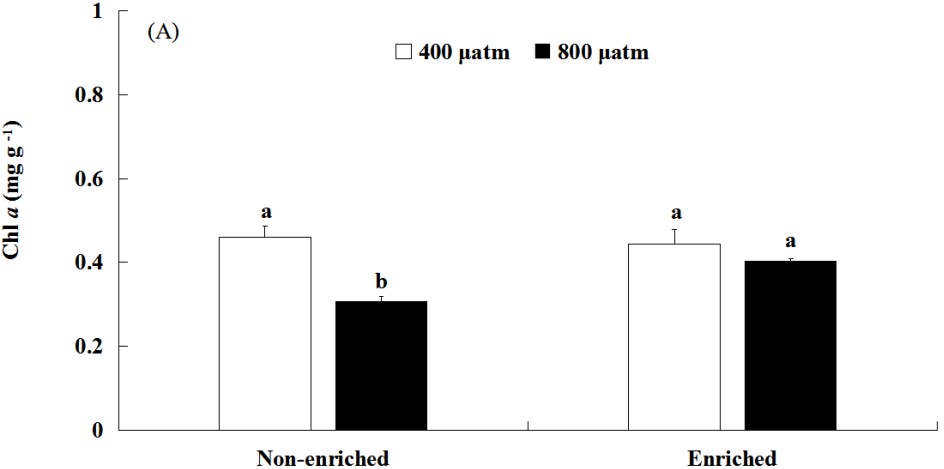

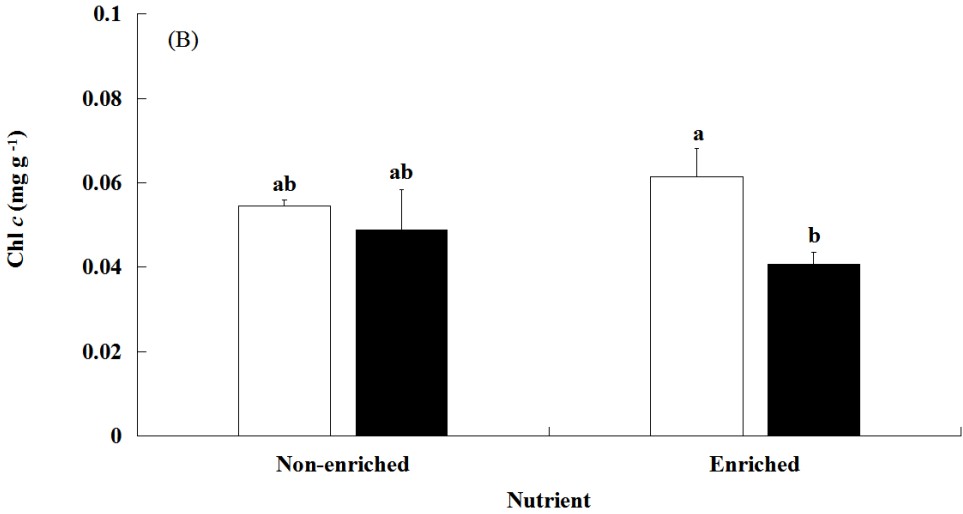

**Figure 3** **The contents of Chl *a* (A) and Chl *c* (B) of *S. japonica* cultured for 6 days under two pCO$_2$ and two nutrient levels.** Data represents mean ± SD (*n* = 3 replicates). Different letters indicate statistical differences (*p* < 0.05) among different experimental treatments.

lower pCO$_2$ and higher pCO$_2$ levels. The Chl *c* content showed a maximum of 0.062 mg g$^{-1}$ at the lower pCO$_2$ level and nutrient-enriched conditions.

## Soluble carbohydrates and soluble proteins

The soluble carbohydrates content was significantly affected by pCO$_2$ (Fig. 4A, Table 2). There was no significant effect of nutrients on the soluble carbohydrates content. In addition, a significant interaction between pCO$_2$ and nutrients was not detected. Under nutrient-enriched conditions, the soluble carbohydrates content at higher pCO$_2$ was significantly lower than that at lower pCO$_2$ levels. However, under non-enriched

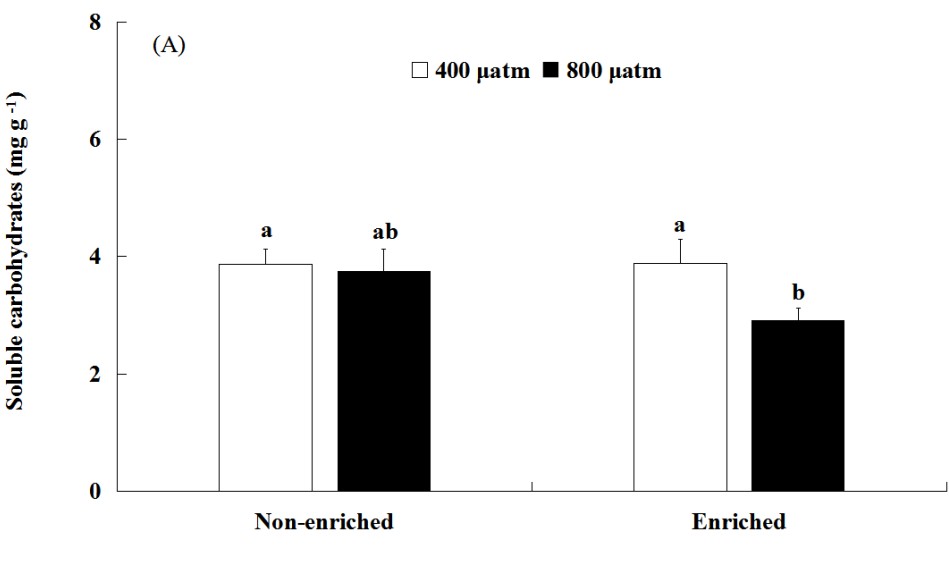

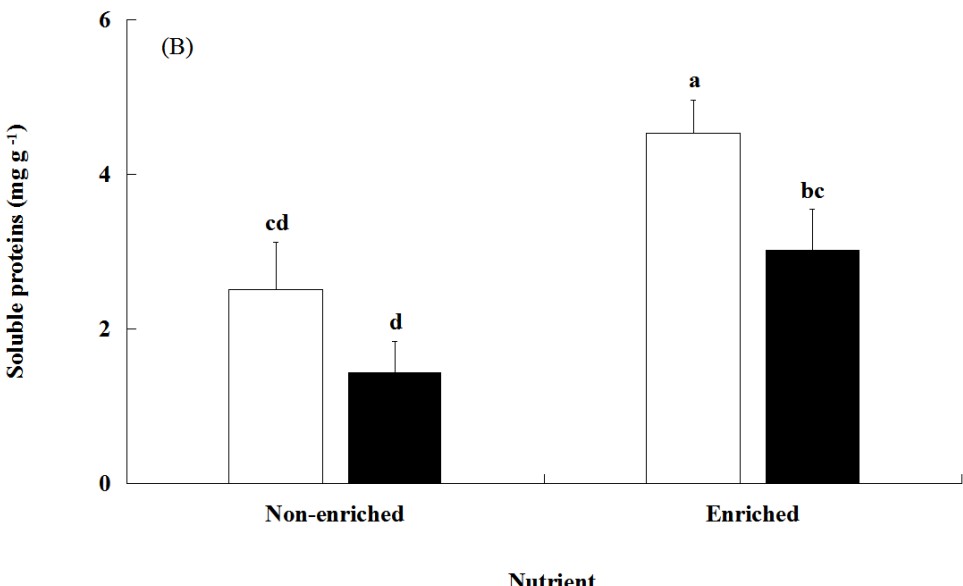

**Figure 4** **The contents of soluble carbohydrates (A) and soluble proteins (B) of *S. japonica* cultured for 6 days under two pCO₂ and two nutrient levels.** Data represents mean $\pm$ SD ($n = 3$ replicates). Different letters indicate statistical differences ($p < 0.05$) among different experimental treatments.

conditions, the soluble carbohydrates content showed no significant differences between the two pCO₂ levels. The soluble carbohydrates content showed a minimum of 2.912 mg g$^{-1}$ at the higher pCO₂ level and nutrient-enriched conditions.

The soluble proteins content was significantly affected by both pCO₂ and nutrients (Fig. 4B; Table 2). However, a significant interaction between pCO₂ and nutrients was not detected. Under non-enriched conditions, the soluble proteins content had no significant differences between the two pCO₂ levels. However, under nutrient-enriched conditions,

the soluble proteins content at high $pCO_2$ was significantly lower than that at lower $pCO_2$. For both $pCO_2$ levels, the soluble proteins content was significantly higher under nutrient-enriched conditions than under non-enriched conditions. The soluble proteins content showed a maximum of 4.536 mg $g^{-1}$ at the lower $pCO_2$ level and nutrient-enriched conditions.

## DISCUSSION

In the present study, the growth of adult sporophytes of *S. japonica* was significantly decreased at the expected future $pCO_2$ level of 800 μatm. Similarly, sporophytes of *Saccharina latissima* and *Fucus vesiculosus* responded to elevated $pCO_2$ levels with a decrease in the growth rate (*Swanson & Fox, 2007*; *Gutow et al., 2014*). Previous studies have shown that decreased seawater pH disturbed the acid–base balance both at the cell surface and within cells (*Flynn et al., 2012*; *Xu, Gao & Xu, 2017*), thereby affecting carbonic anhydrase activity and the uptake and assimilation of nitrogen (*García-Sânchez, Fernândez & Niell, 1994*; *Israel et al., 1999*). Additionally, macroalgae could enhance the energetic cost to maintain intracellular pH stability (*Xu, Gao & Xu, 2017*), suggesting that the energy used for supporting growth may be reduced. These findings may partially explain why the growth of *S. japonica* decreased under OA conditions. On the other hand, the net photosynthetic rate and chlorophyll content were reduced under the high $pCO_2$ levels. Similar results have been shown for *Alaria esculenta* (*Gordillo et al., 2015*) and *Lomentaria australis* (*Van der Loos, Schmid & Leal, 2019*). Elevated $pCO_2$ could accelerate the degradation of chlorophyll synthesis immoderately, which is not essential for light harvesting (*Gordillo et al., 1998*). *Lapointe & Duke (1984)* also indicated that the photosynthetic activities of macroalgae were closely associated with the intracellular chlorophyll content. The reduced chlorophyll content may have contributed to a decrease in the photosynthetic rate. Additionally, according to previous study, the carotenoid contents were significantly affected by elevated $pCO_2$ (*Celis-Plá et al., 2017*), which could influence the growth and quality of seaweed. However, in the present study, due to the limiting of experimental conditions, the concentration of carotenoid was not determined. But, we will add this parameter in future studies to investigate the effect of elevated $pCO_2$ level on the growth and quality of *S. japonica*.

OA also had a negative effect on soluble carbohydrates and soluble proteins. The negative effect may be partially due to the inhibition of the enzyme activity regarding carbon assimilation due to an elevated $pCO_2$ level (*García-Sânchez, Fernândez & Niell, 1994*), resulting in the reduction of carbohydrates synthesis (*Giordano, Beardall & Raven, 2005*). In *Fucus serratus* (*Axelsson, Uusitalo & Ryberg, 1991*), *Desmarestia aculeata* (*Gordillo et al., 2016*), and *Ptilota plumosa* (*Gordillo et al., 2016*), negative effects of OA on the soluble carbohydrates and soluble proteins were also found. The decreased soluble proteins content was likely due to a decrease in the uptake and assimilation of nitrate (*Lara et al., 1987*; *Mercado et al., 1999*), which is used for the synthesis of structural proteins involved in algal growth (*Chen, Zou & Yang, 2017*). On the other hand, the exposure and acclimation to elevated $pCO_2$ levels would influence the reallocation of nitrogen away from Rubisco

and towards other limiting components, such as non-photosynthetic processes (*Bowes, 1991*; *Andria, Vergara & Perez-Llorens, 1999*).

Significant positive effects of high nutrient availability were detected for Chl *a* content, soluble proteins content and growth. Similarly, in *U. pinnatifida* (*Endo et al., 2017*), the growth rate and Chl *a* content were higher in the 25% PESI-enriched treatment than in the non-enriched treatment. Stimulation of soluble proteins synthesis has also been reported in *Ulva rigida* (*Gordillo, Niell & Figueroa, 2001*). A higher nitrogen availability may promote the synthesis of Chl *a* and related enzymes (*Dawes & Koch, 1990*; *Crawford, 1995*). Increases in these parameters stimulated growth under high nutrient conditions and further enhanced the recruitment and natural production of kelp (*Mizuta, Narumi & Yamamoto, 2001*; *Agatsuma et al., 2014*). However, soluble carbohydrates were not significantly affected by a high nutrient supply. This result suggests that the synthesis of soluble carbohydrates was less sensitive than the other physiological parameters to changes in the nutrient supply; in future studies, the responses of multiple physiological parameters to changes in the nutrient supply should be investigated to explain these different phenomena.

Adult sporophyte growth was more severely decreased by a combination of elevated $pCO_2$ and nutrient levels than by an increased $pCO_2$ level individually. The results implied that natural and cultivated production of *S. japonica* will be inhibited in the future when both nutrient and $pCO_2$ (800 $\mu$atm) levels are higher than the current natural seawater conditions. However, according to previous studies, the growth of adult sporophytes of *U. rigida* was reduced due to reproduction that resulted in a loss of thallus mass under high $pCO_2$ and nutrient conditions (*Gao et al., 2017a*; *Gao et al., 2017b*; *Gao et al., 2018b*). In the present study, we did not observe a large amount of spore release. The decreased growth of adult *S. japonica* under elevated $pCO_2$ and nutrient conditions may be a simple stress response to OA. In addition, high nutrient availability alleviated the negative effect of OA on the growth of *Pyropia yezoensis* (*Gao et al., 2019c*). The divergence of this species may be because elevated $pCO_2$ induced the synthesis of functional proteins under higher nutrient conditions to scavenge reactive oxygen species and protect cells from harm caused by decreased pH (*Gao et al., 2019c*). According to these differential results, the interactions of elevated $pCO_2$ and nutrients on seaweeds are algal species-specific. The phenomenon was possibly caused by the differences in their response mechanisms and the ability of acclimation under high $pCO_2$ and nutrient conditions.

In total, climate changes present a great challenge for mariculture production of commercial macroalgae. As an environmental factor, OA could produce a negative effect on the production and quality of *S. japonica*. Continuous eutrophication aggravates deterioration, a negative effect of OA. However, due to limited physiological data in this study, more experiments regarding the responses of different developmental stages to OA and eutrophication are warranted to evaluate the development of *S. japonica* cultivation in future oceanic conditions with high $pCO_2$ and nutrient levels.

## CONCLUSIONS

In this study, the combined effects of OA and eutrophication on the growth and biochemical compositions of adult sporophytes of *S. japonica* were investigated. The results showed that OA showed a significantly negative effect on growth, photosynthesis, chlorophyll, and soluble carbohydrates and proteins. Moreover, eutrophication exacerbated the negative effect of OA on growth. These results suggested that the cultivation production and commercial value of this kelp would be reduced under the future ocean conditions.

## ACKNOWLEDGEMENTS

We would like to thank anonymous reviewers for helpful comments that improved the manuscript.

### Funding
The authors received no funding for this work.

### Competing Interests
The authors declare there are no competing interests.

### Author Contributions
- Yaoyao Chu conceived and designed the experiments, performed the experiments, analyzed the data, prepared figures and/or tables, approved the final draft.
- Yan Liu, Jingyu Li and Qingli Gong conceived and designed the experiments, contributed reagents/materials/analysis tools, authored or reviewed drafts of the paper, approved the final draft.

### Data Availability
  The raw measurements are available in the Supplemental File.

### Supplemental Information
Supplemental information for this article can be found online at http://dx.doi.org/10.7717/peerj.8040#supplemental-information.

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
