# Peer review of "Effects of elevated pCO2 and nutrient enrichment on the growth, photosynthesis, and biochemical compositions of the brown alga Saccharina japonica (Laminariaceae, Phaeophyta)"

_PeerJ, doi:10.7717/peerj.8040_

## Round 0.1 · original submission · Major Revisions

Please see the Reviewers' comments below and revise your manuscript accordingly. Two Reviewers have pointed out issues with language. Please proofread the manuscript and correct any language-related issues, in addition to editing the scientific content.

Reviewer 1 ·

Basic reporting

Good

Experimental design

Good

Validity of the findings

Good

Additional comments

This manuscript investigated the combined effects of two critical environmental factors, ocean acidification (OA) and eutrophication on physiological performances in the economy important macroalga Saccharina japonica. The authors measured relative growth rate (RGR), net photosynthetic rate, and biochemical compositions (including chlorophyll (Chl) a, Chl c, soluble carbohydrates, and soluble protein) and found the negative effect of OA and interaction of OA and eutrophication on some key parameters. This study provides useful understanding on how production and chemical composition of S. japonica would be affected in the future oceans. The experiments were well designed and performed. The data analyses were satisfactory. I thus think this manuscript can be published in Peer J after revision.

Specific comments
Line 22 For pCO2, the unit of uatm rather than ppmv is usually used. Please correct it throughout the text.
Line 25 400 ppmv of what?
Line 29 Aggravate is not a proper word here.
Line 34 The section of Introduction is well written.
Line 92 How quickly?
Line 102 Culture period should be pointed out in this section
Line 104 the concentration of nitrate and phosphate in the two nutrient levels should be presented.
Line 113 Please provide the mode, brand and product place for the two incubators

Discussion section
I am wondering whether the combination of OA and eutrophication led to reproduction events, which caused the decrease of growth, particularly after fragmentation (1.4 cm in diameter). This could happen in Ulva species (Gao et al., 2017ab, 2018). Please discuss it in the text.
Gao G, Clare AS, Rose C, Caldwell GS. Eutrophication and warming-driven green tides (Ulva rigida) are predicted to increase under future climate change scenarios. Marine Pollution Bulletin, 2017a, 114:439-447.
Gao G, Clare A S, Rose C, Caldwell GS. Intrinsic and extrinsic control of reproduction in the green tide-forming alga, Ulva rigida. Environmental and Experimental Botany, 2017b, 139: 14-22.
Gao G, Clare AS, Rose C, Caldwell GS. Ulva rigida in the future ocean: potential for carbon capture, bioremediation, and biomethane production. Global Change Biology Bioenergy, 2018, 10: 39-51.

Line 279 Positive interaction sounds weird.
Line 296 I cannot see “eutrophication exacerbated the negative effect of OA on growth” based on Figure 1. Please describe the interaction in the Results section.
Line 514 Correct 2012 to 2017.
Figure 1 Add space between number and letter. Correct it throughout the figures.
Figure 2 The unit of net photosynthetic rate is not correct.
Table 2 Italicize a and c for chlorophyll

Reviewer 2 ·

Basic reporting

no comment

Experimental design

The experiment was well designed.

Validity of the findings

no comment

Additional comments

This study focuses on the effects of pCO2 and nutrient enrichment on growth, photosynthesis, and biochemical compositions of the brown alga Saccharina japonica (Laminariaceae, Phaeophyta). The paper is well written and the results are clearly presented. but a number of points need to be addressed. The language needs considerable help by a native speaker. Sentence constructions are awkward throughout the whole manuscript leading to major misunderstanding or non-understanding of the text passages.

Introduction: Hypothesis missing.

The whole growth experiment last for only 6 days, why did you do the experiment for longer time because the growth rate of this brown alga was less than 2%?

Please provide the information about nutrient condition at nature seawater and enriched seawater.

Figure 2, μmol.

The unit of PCO2 in seawater is μatm, not ppmv

Table 1, please provide the unit of the parameters of seawater carbonate system.

References: Xu Z, Gao G, Xu J. 2017. Physiological response of a golden tide alga (Sargassum muticum) to 515 the interaction of ocean acidification and phosphorus enrichment. Biogeosciences 14: 671-681.

Reviewer 3 ·

Basic reporting

This study showed the changes in the growth, photosynthesis, and biochemical compositions of the brown alga Saccharina japonica under elevated pCO2 and nutrient enrichment.

1. It’s not clear why the authors didn’t determine the concentration of carotenoid, which is important photosynthetic pigment as important as Chl a and Chl c for the Saccharina japonica.

2. One would strongly disagree with claim that “Despite the important effects of OA and eutrophication on kelp species, very few studies have considered the combined effects of these factors on the physiological characteristics of kelp species”.

I suggest the authors to well search the literatures. At least, there are several cited in this manuscript already:
Gao G, Gao Q, Bao M, Xu J, Li X. 2019. Nitrogen availability modulates the effects of ocean acidification on biomass yield and food quality of a marine crop, Pyropia yezoensis. Food Chemistry 271:623-629 DOI 10.1016/j.foodchem.2018.07.090.

Gao G, Beardall J, Bao M, Wang C, Ren W, Xu J. 2018. Ocean acidification and nutrient limitation synergistically reduce growth and photosynthetic performances of a green tide alga Ulva linza. Biogeosciences 15(11):3409-3420 DOI 10.5194/bg-15-3409-2018.

Experimental design

1. What are the nutrients and their concentrations in the 50% PESI enriched seawater? The authors need to explain why this level was set up. This is key information for the whole study.

2. Along with my concern on the nutrient level used in this study, a further question is what about the nutrient scenarios in the previous studies on the responses of seaweed to OA and eutrophication? This may also help explain the different results observed in these studies. Now the authors simply conclude that “These results indicate that the interactions of elevated pCO2 and nutrient on seaweeds are algal species-specific.” A better approach could be setting up a series of nutrients, considering that the CO2 is only 1 treatment and 1 control group (air).

3. The concentration of carotenoid is recommended to be detected in such a study.

Validity of the findings

The authors need to expand the statements for their novelty.

Additional comments

English writing need to be improved; and the authors need to check throughly for the minor issues.

For instance, L65-66: “The canopy-forming kelp Saccharina japonica (Areschoug) Lane, Mays, Druehl and Saunders inhabits subtidal zones”, what’s the Lane, Mays, Druehl and Saunders?

---

## Round 0.2 · Minor Revisions

As you can see below, Reviewer 3 had some additional comments on your revised manuscript. Pease see if you can address these concerns in the manuscript and provide a response letter explaining the changes. In case no changes will be made, please explain the reasons for not changing the manuscript.

Reviewer 1 ·

Basic reporting

No comment.

Experimental design

No comment.

Validity of the findings

No comment.

Additional comments

The authors have revised the manuscript based on my comments. I thus suggest accepting this manuscrip.

Reviewer 3 ·

Basic reporting

The basic written English has been improved throughout the manuscript.

Experimental design

To keep the readers aware of the major flaw in their experiment design. The authors need to add explanations in the methods and discussions that, they didn't determine the concentration of carotenoid, which is important photosynthetic pigment as important as Chl a and Chl c for the Saccharina japonica.

The authors need also consider my previous comment: "A better approach could be setting up a series of nutrients, considering that the CO2 is only 1 treatment and 1 control group (air)."

Validity of the findings

no further comment

---

## Round 0.3 · accepted · Accept

It was a pleasure working with you. I am looking forward to the publication of your article and wish you continued success with your future projects.

Reviewer 3 ·

Basic reporting

I have no further comments

Experimental design

I have no further comments

Validity of the findings

I have no further comments

Additional comments

I have no further comments